

# Quantifying the glacial meltwater contribution to streams in mountainous regions using highly resolved stable water isotope measurements

Philipp Wanner[1], Noemi Buri[2], Kevin Wyss[2], Andreas Zischg[2], Rolf Weingartner[2], Jan Baumgartner[3], Benjamin Berger[3], Christoph Wanner[4]

[1]Department of Earth Sciences, University of Gothenburg, Guldhedsgatan 5A, 413 20 Gothenburg, Sweden
[2]Institute of Geography, University of Bern, Hallerstrasse 12, 3012 Bern, Switzerland
[3]Kraftwerke Oberhasli AG (KWO), Grimselstrasse 19, 3862 Innertkirchen, Switzerland
[4]Institute of Geological Sciences, University of Bern, Baltzerstrasse 1+3, 3012 Bern

*Correspondence to*: Philipp Wanner (+46 76 618 56 15; philipp.wanner@gu.se)

**Abstract.** This study aims to determine the contribution of glacial meltwater to streams in mountainous regions based on stable water isotope measurements ($\delta^{18}O$ and $\delta^{2}H$). For this purpose, three partially glaciated catchments were selected as the study area in the central Swiss Alps being representative of catchments that are used for hydropower energy production in Alpine regions. The glacial meltwater contribution to the catchments' stream discharges was evaluated based on high-resolution $\delta^{18}O$ and $\delta^{2}H$ measurements of the end-members that contribute to the stream discharge (ice, rain, snow) and of the discharging streams. The glacial meltwater contribution to the stream discharges could be unequivocally quantified after the snowmelt in August and September when most of the annual glacial meltwater discharge occurs. In August and September, the glacial meltwater contribution to the stream discharges corresponds to up to 95±2% and to 28.7%±5% of the total annual discharge in the evaluated catchments. The high glacial meltwater contribution demonstrates that the mountainous stream discharges in August and September will probably strongly decrease in the future due to global warming-induced deglaciation, which will be, however, likely compensated by higher discharge rates in winter and spring. Nevertheless, the changing mountainous streamflow regimes in the future will pose a challenge for hydropower energy production in the mountainous areas. Overall, this study provides a successful example of an Alpine catchment monitoring strategy to quantify the glacial meltwater contribution to stream discharges based on stable isotope water data, which leads to a better validation of existing modelling studies and which can be adapted to other mountainous regions.

## 1. Introduction

The current global energy production still heavily relies on fossil energy resources such as oil and gas, emitting large quantities of greenhouses gases such as $CO_2$ and methane into the atmosphere. Currently, 53 gigatons tons of greenhouse gases are annually released to the atmosphere significantly contributing to continuous global warming (IPCC, 2018). To decelerate climate change, the atmospheric emission of greenhouse gases needs to be drastically reduced, whereby a greenhouse gas emission reduction of 80% until 2030 and of 100% until 2050 is necessary to avoid a global warming of 1.5°C (Jacobsen and Hjelmso, 2014; IPCC, 2018). For achieving these greenhouse emission reduction goals, energy production from carbon neutral renewable energy resources such as hydropower, wind, wave, solar and geothermal systems play an important role. Among renewable energy technologies, hydropower is currently the most important resource accounting for 72% of the global renewable electricity and for 16% of the total global electricity production (Gernaat et al., 2017). A large portion of hydropower is produced in mountainous areas using water bodies in artificially dammed lakes, especially in Alpine regions, where this type of hydropower represents about 60% of the total electricity production (Schaefli et al., 2019). However, artificially dammed lake reservoirs rely on the contribution from water resources that are temporarily stored in glacial ice being sensitive to global warming (Barnett et al., 2005). There is a broad agreement that the relative contribution of rain, snow melt, and





glacial meltwater to artificially dammed lakes via mountainous stream discharges will change due to global warming and
impact hydropower production (Bolch et al., 2012; Bombelli et al., 2019; Bradley et al., 2006; D'agata et al., 2018; Finger et
al., 2012; Orlove, 2009; Patro et al., 2018; Puspitarini et al., 2020). To evaluate the effect of climate change on hydropower
production using mountainous streams and artificially dammed lakes, it is of major importance to gain knowledge about the
different mountainous stream components. In particular, it is crucial to quantify the relative contribution of glacial meltwater
to mountainous streams since this component will likely disappear in the future caused by global warming. These investigations
are important to develop future strategies for ensuring a continuous hydropower energy production in the course of global
warming. The continuous operation of hydropower plants under changing climate conditions is particularly important for
achieving the $CO_2$ emission reduction goals as hydropower is the most importance renewable carbon neutral energy resource.
Up to present several methodological approaches have been used to quantify the relative contribution glacial melt, snowmelt
and rain water to the total discharge of mountainous streams including direct discharge measurements, glaciological methods,
hydrological balance equations, hydro-chemical tracers, and hydrological modelling (Frenierre and Mark, 2014). It has been
demonstrated that the hydro-chemical tracer method has several advantages compared to other methods including a low
dependency on existing data, the possibility to capture temporal and spatial variations of the different contributions to total
discharge, the applicability from micro- to mesoscale and the low costs (Frenierre and Mark, 2014). The hydro-chemical tracer
method relies on the different hydro-chemical signatures such as stable oxygen and hydrogen isotope ratios, electrical
conductivity, ion concentration and temperature of waters that originate from various "end-members" including rain, snow
and ice. The distinct hydro-chemical signatures open-up the possibility to quantify the proportion of these endmembers in
streamflow. The hydro-chemical tracer approach was applied by a number of studies in South America, Asia and India (Boral
et al., 2019; Laskar et al., 2018; Lone et al., 2017; Mark and Mckenzie, 2007; Ohlanders et al., 2013) for quantifying the
contribution of glacial meltwater to mountainous stream discharge. In contrast to South America, Asia and India modelling
approaches were primarily used in Alpine regions to estimate the contribution of glacial meltwater to streams and to evaluate
how this glacial meltwater contribution is impacted by climate change (Bombelli et al., 2019; D'agata et al., 2018; Finger et
al., 2012; Patro et al., 2018; Schaefli et al., 2019; Puspitarini et al., 2020). However, these modelling approaches are strongly
dependent on assumptions, existing data sets and are often related with significant uncertainties. Hence, they are often not
providing an accurate quantification of the different contributors including rain, snow melt, glacial melt water to mountainous
streams.

67          This study aims to apply the hydro-chemical tracer method in Alpine regions to quantify the glacial meltwater

contribution to mountainous streams with a lower uncertainty compared to the previously conducted modelling studies. For
that purpose, three partially glaciated Alpine watersheds were selected in the central Alpine region in Switzerland, where
mountainous streams are used for hydropower energy production. To quantify the contribution of glacial meltwater to the
streamflow in the three mountainous catchments areas, highly temporally resolved stable water isotope ($\delta^{18}O$ and $\delta^2H$),
electrical conductivity and discharge measurement were conducted between July 2019 and March 2020. The measurements
provide detailed insight into glacial and snow melt processes and their influence to mountainous stream discharges in Alpine
regions. Moreover, this study provides excellent information on the continuous monitoring of mountainous catchments for the
quantification of the glacial meltwater contribution to mountainous streams using stable water isotopes leading to a better
validation of the available modelling studies.
**2. Materials and Methods**
**2.1 Site description**
The three selected catchment areas are located in the Gadmen valley in the Central Swiss Alps and are named Steinwasser,
Giglibach and Wendenwasser (Fig. 1). The three catchments are located in the Aar massif, consisting of metamorphic Gneiss





and Granites that are overlain by moraine and talus material with various thicknesses. The three catchments are located adjacent
to each other (Fig. 1) and have a similar average elevation ranging between 2'190 and 2'471 meters above sea level (masl).
The catchment areas differ in their size and degree of glaciation, whereby the Steinwasser catchment shows the largest size
and degree of glaciation (24.2 km$^2$; 28.0%), followed by the Wendenwasser (11.2 km$^2$; 14.9%) and Giglibach catchment (4.9
km$^2$; 6.0%). The difference in sizes and degrees of glaciation of the three catchment areas provide the advantage that the hydro-
chemical tracer method can be applied under various conditions to quantify the glacial meltwater contribution to the stream
discharges.

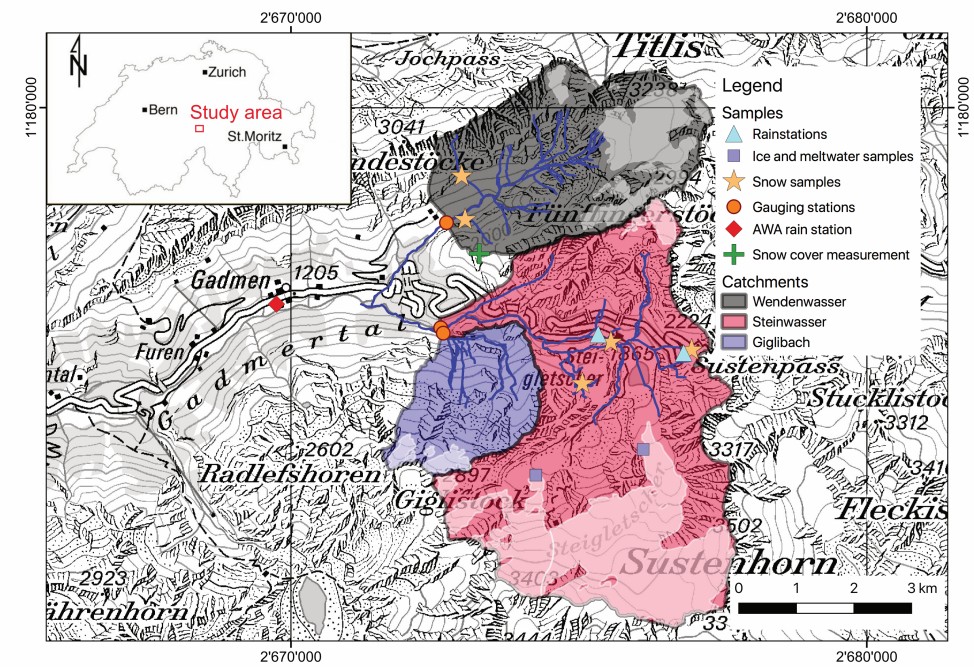

**Figure 1: Areas of the Wendenwasser (pink), Steinwasser (grey) and Giglibach (blue) catchment, which are located in the Gadmen**
**valley in the central part of the Swiss alps and the sampling locations of the ice (pink squares), snow (orange stars) and rain (turquoise**
**triangles) end-members as well as the catchment effluent measuring stations (orange circles). The red diamond represents the AWA**
**rain station, whereas the green cross indicates the location of the Gschletteregg snow measuring station, and the white areas show**
**the glaciated parts of the three catchments**
**2.2. Field measurements and sampling**
**2.2.1 End-member sampling**
To characterize the hydro-chemical signatures of the end-members (rain, ice, snow) that contribute to the streams in the three
catchment areas and to analyze their potential spatio-temporal evolution, each end-member was sampled and analyzed several
times at various locations in the three catchment areas (Fig. 1). The rain end-member was sampled by using a 1B Palmex rain
collector at the merging effluents of the Steinwasser and Giglibach catchment (1'430 masl) and a Young rain collector (Nr.
55203) at the effluent of the Wendenwasser catchment (1'542 masl; Fig. 1). In addition, two Young rain collectors (Nr. 55203)
were installed in the Steinwasser catchment at two different altitudes (1'842 and 2'210 masl) to capture potential changes of
the $\delta^{18}O$ and $\delta^2H$ rain signal as a function of the altitude. The rain was sampled from the rain collectors in 16 days intervals
between July 2019 and October 2019 and stored in 300 mL plastic bottles prior to analysis. Besides, precipitation data was
acquired from June 2019 to March 2020 from the precipitation measuring station in Gadmen (Fig. 1) operated by the



Office of Water and Waste (AWA) of the Canton of Bern, Switzerland (AWA, 2021). To determine the hydro-chemical
signature of the snow end-member a high number of snow samples (19) were taken at various locations, at different elevations
and at different times between February 2019 and March, 2020 in the three catchment areas to capture the spatio-temporal
hydro-chemical variation of the snow (Fig. 1). At each location, the snow was sampled vertically from the snow surface to the
bottom of the snow cover using a Standard Federal Snow Tuber (SFST). After sampling, the snow was transferred into a wide
mouth PET bottle, which was closed immediately after filling to ensure that the snow melts in a closed container to avoid an
evaporation-induced alteration of the sample prior to laboratory analysis. In addition to the taken snow samples, the thickness
of the snow cover was measured every ten minutes at the Gschletteregg measuring station at 2'063 masl (Fig. 3) being operated
by the Swiss Institute for Snow and Avalanche Research (SLF). For measuring the hydro-chemical signature of the ice-end-
member, ice samples were taken from the glaciated areas in the three catchments using an ice pick between August 2019 and
September 2019 (Fig. 1). To ensure that the taken ice samples are representative for the meltwater component in the streams,
solid ice as well as melting ice samples were taken from the ablation zone of the glaciated area, whereby the uppermost
centimeters were scraped off and not used during sampling. Similar to the snow samples, the sampled ice was transferred into
a plastic container and afterwards immediately closed that no alteration of the ice samples occurred during melting before it
was analyzed in the laboratory.
**2.2.2 Field station sampling and measurements**
To measure the discharges and the electrical conductivity in three catchments' effluents and for taking stream samples for
stable water isotope analysis, three field measuring stations were deployed at the effluents of the three catchments (Fig. 1).
The measurements were conducted between June 2019 and March 2020, whereby the exact sampling period for each parameter
and catchment effluent is provided in table 1.

Table 1. Sampling periods for different parameters in catchment effluents

| Catchment | Parameter | Sampling period |
|---|---|---|
| Wendenwasser | Discharge | July 31, 2019 - March 21, 2020 |
| Wendenwasser | Electrical conductivity | August 13, 2019 – November 7, 2019 |
| Wendenwasser | Stable water isotopes ($\delta^{18}$O, $\delta^{2}$H) | July 31, 2019 - March 9, 2020 |
| Steinwasser | Discharge | June 19, 2019 – March 2, 2020 |
| Steinwasser | Electrical conductivity | June 19, 2019 – February 29, 2020 |
| Steinwasser | Stable water isotopes ($\delta^{18}$O, $\delta^{2}$H) | June 18, 2019 – March 9, 2020 |
| Giglibach | Discharge | July 17, 2019 – March 21, 2020 |
| Giglibach | Electrical conductivity | July 17, 2019 – March 9, 2020 |
| Giglibach | Stable water isotopes ($\delta^{18}$O, $\delta^{2}$H) | July 17, 2019 – March 9, 2020 |


The stream discharges and the electrical conductivity were measured every 10 seconds and averaged over a 10 minutes interval
during the monitoring periods. The stream discharges were determined via stream level measurements using the
discharge/water level (P/Q) relationship, whereby the P/Q relation in the three streams were determined using the salt dilution
method at various stream water levels (Wyss, 2020). The electrical conductivity of the streams was measured using a Campbell
Scientific probe. Samples for stable water isotope analysis (oxygen and hydrogen) were taken from the catchments' effluents
using an autosampler design from the University of Freiburg, Germany between June and October 2019. To avoid evaporation
between stream water sampling and analysis, 180 drops of Paraffin was added to the empty sample bottles prior to stream
sampling as conducted by previous studies (Michelsen et al., 2018; Ohlanders et al., 2013). The water-insoluble Paraffin
remains on top of the water during sampling due to its lower density compared to water preventing the evaporation and



alteration of the sample. After October 2019, samples for stable water isotope analysis were taken manually since the increasing
snow cover prevented the continuous automatic monitoring by using the autosampler.

**2.3 Laboratory analysis**

**2.3.1 Stable water oxygen and hydrogen isotope measurements of end-members and stream**

The stable oxygen and hydrogen isotope ratios of rain, snow, ice and stream discharge samples were analysed using a Picarro
L2120-I cavity ring down spectrometer (CRDS) with vaporization module V1102-I at the Institute of Geological Science,
University of Bern, Switzerland. The measured stable oxygen and hydrogen isotope ratios were expressed in the delta notation
($\delta$ = (R/RStd - 1) · 1000 (‰)), where R and $R_{Std}$ are the isotope ratios of the sample and the standard, respectively. Raw $\delta^{18}O$
and $\delta^2H$ values are obtained by a tenfold measurement of each sample followed by a post run-correction (memory and drift)
according to van Geldern and Barth (2012). To obtain $\delta^{18}O$ and $\delta^2H$ values on the international Vienna Standard Mean Ocean
Water (VSMOW) scale, raw delta values were calibrated against two internal standards, which were referenced to the VSMOW
scale using international IAEA standards. The two standards used for calibration differed in their isotope composition and span
a calibration interval between -27.41‰ and -2.65‰ for $\delta^{18}O$ values and between -209.8‰ and -13.9‰ for the $\delta^2H$ values,
respectively. The analytical uncertainty of the $\delta^{18}O$ and $\delta^2H$ measurements was determined based on multiple internal and
IAEA standard analysis and corresponds to 0.10‰ and 1.5‰, respectively.

**2.4 Discharge separation based on stable isotope measurements.**

The contribution of the different end-members (ice, rain, snow) to the discharges of the three catchments was quantified based
on highly resolved stable water isotope ratio ($\delta^{18}O$, $\delta^2H$) measurements in the catchments' effluents. To quantify the end-
member discharge contribution, the two end-members were considered that contributed predominately to the discharge using
a binary mixing approach:
$I_{Effluent} = X \cdot I_{End-member1} + (1-X) \cdot I_{End-member2}$ (1)
where $I_{Effluent}$ is the isotopic composition of the catchment's effluent, $I_{End-member1}$ and $I_{End-member2}$ are the isotopic compositions of
the end-members (snow, rain, ice) and X is the contribution of the end-members to the effluent.
To quantify the contribution of each end-member to the catchment's effluent, equation 1 was resolved to X:
$X = (I_{Effluent} - I_{End-member2})/(I_{End-member1} - I_{End-member2})$ (2)

**3. Results and Discussion**

**3.1 Analysis of isotopic composition of end-members**

The temporal oxygen and hydrogen isotope evolutions ($\delta^{18}O$ and $\delta^2H$) of the three end-members that were considered in this
study (rain, snow, ice) are illustrated in Figure 2. The rain end-member showed similar average $\delta^{18}O$ and $\delta^2H$ values at the
effluent location of the Wendenwasser catchments (-8.14‰; -54.1‰) and at the merging effluent location of the Giglibach
and Steinwasser catchment (-8.24‰; -52.4‰), respectively (Fig. 1) during the entire rain sampling period (August 8 – October
18). Compared to the effluent locations, the rain was more depleted in $^{18}O$ and $^2H$ at higher altitude at the low and high
Steinwasser rain sampling location (Fig. 1) during the early stage of the sampling period (August 8 – 29, 2019) showing a $\delta^{18}O$
shift of 0.27‰/100m and a $\delta^2H$ shift of 1.8‰/100m, respectively. The depletion of $^{18}O$ and $^2H$ in the rain with increasing





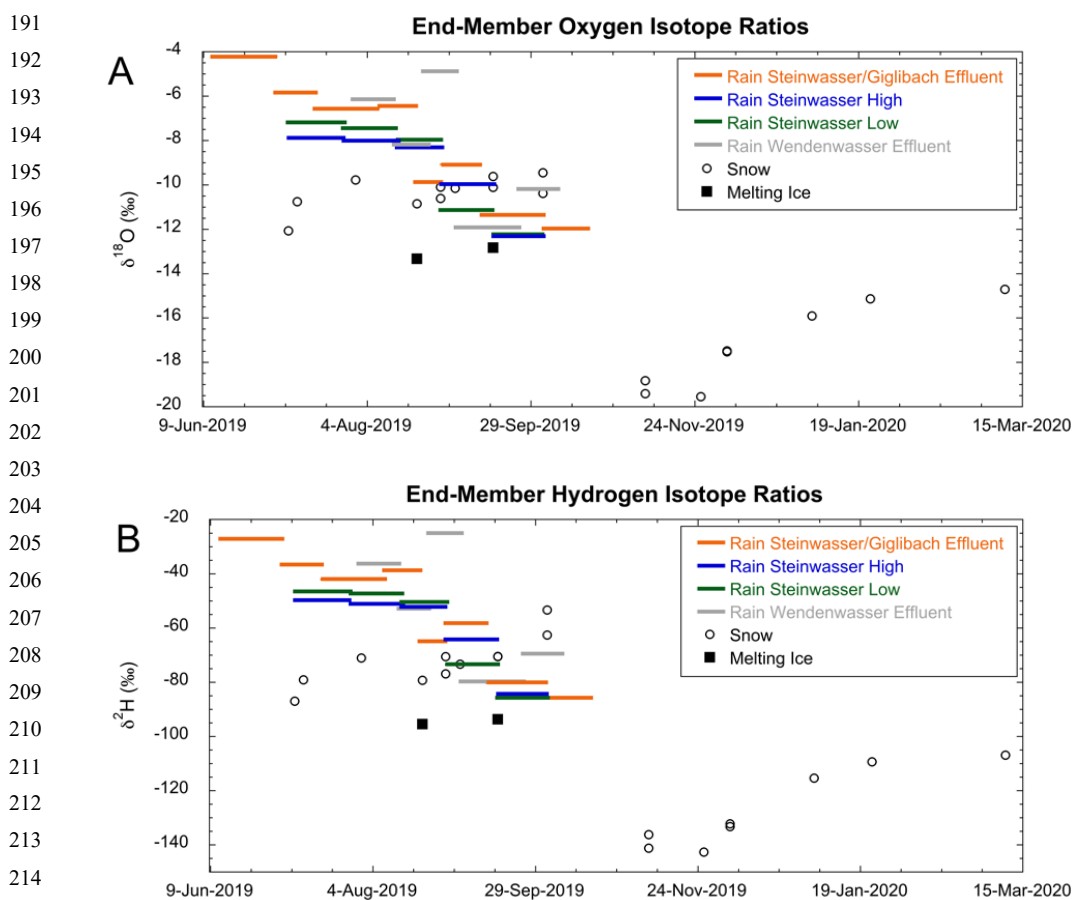

**Figure 2: Temporal δ¹⁸O (A) and δ²H (B) evolution of the end-members including rain (solid lines), snow (open circles) and melting ice (filled squares) during the sampling period (June 2019 until March 2020) in the Steinwasser, Wendenwasser and Giglibach catchment.**

altitude can be attributed to the altitude isotope effect, which includes the preferential precipitation of heavy isotopes during the continuous orogenic uplift of humid air masses (Clark and Fritz, 1997). However, in contrast to the early stage of the sampling period, no altitude isotope effect was observed during the later stage of the sampling period (August 30 – October 3), which might result from different meteorological conditions such that no continuous orogenic uplift of the humid air masses and precipitation occurred. Consequently, a continuous depletion of heavy isotopes with increasing altitude could not be observed for the entire sampling period and no overall δ¹⁸O and δ²H altitude correction factor could be established for the three catchments. As opposed to the ambiguous δ¹⁸O and δ²H variations as a function of the altitude, a more distinct temporal δ¹⁸O and δ²H evolution was observed in the three catchments during the sampling period. While the rain was enriched in ¹⁸O and ²H in June 2019, (δ¹⁸O = -4.19‰; δ²H = -26.9‰) it became progressively lighter with increasing time reaching delta values of δ¹⁸O = -12.26‰ and δ²H = -84.3‰, respectively in October 2019 (Fig. 2). This progressive depletion of ¹⁸O and ²H over time can be associated with the seasonal changes and the accompanying temperature decrease between June and October (Clark and Fritz, 1997). To characterize the snow endmember in the three catchments areas, the stable oxygen and hydrogen isotope ratios of the snow was measured during the snow accumulation and ablation period, respectively as the isotopic signal of snow can differ significantly during these two periods (Beria et al., 2018; Cooper, 1998; Dietermann and Weiler, 2013; Lee et al., 2010; Zhou et al., 2008). Based on the monitoring of the snow thickness at the Gschletteregg measuring station (Fig. 1),





the snow accumulation period during which the snow becomes progressively thicker was identified from early November until
April, whereas the snow ablation period during which the snow cover becomes continuously thinner due to sublimation,
melting and redistribution processes was observed between May and October (Fig. 3).


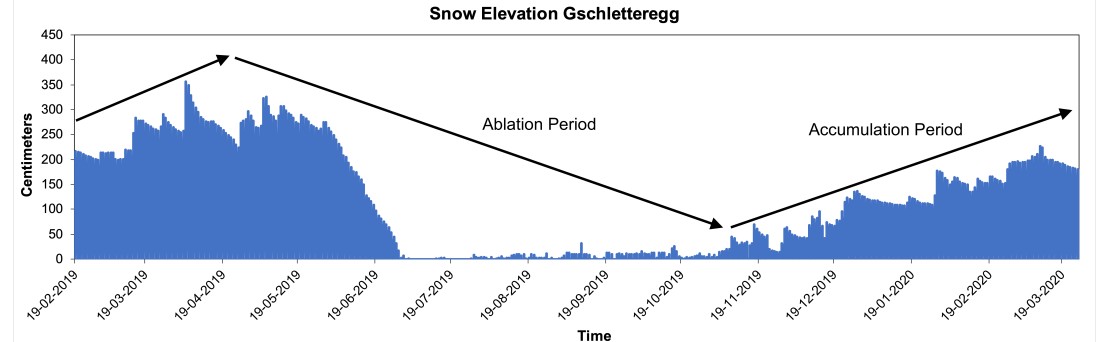

**Figure 3: The measured snow thickness over time at the Gschletteregg measuring station being representative for the three**
**catchment areas. Between November and April snow accumulation occurs, while from May to October snow ablation takes place.**

During the snow accumulation period (November to April) the snow samples revealed average $\delta^{18}O$ and $\delta^2H$ values of -
17.31‰ and -127.1‰, respectively (Fig. 2). The lowest $\delta^{18}O$ and $\delta^2H$ values (-19.40‰; -141.2‰) were detected in November
at the beginning of the snow accumulation period. With increasing time, a continuous enrichment of $^{18}O$ and $^2H$ isotopes was
observed in the snow reaching $\delta^{18}O$ and $\delta^2H$ snow values of -14.70‰ and -106.9‰, respectively, at the end of the accumulation
period in April. The progressive enrichment of $^{18}O$ and $^2H$ in the snow occurred along the LMWL (Fig. 4) and hence, is not
explainable by sublimation processes, which would lead to a righthand deviation from the LMWL (Beria et al., 2018).


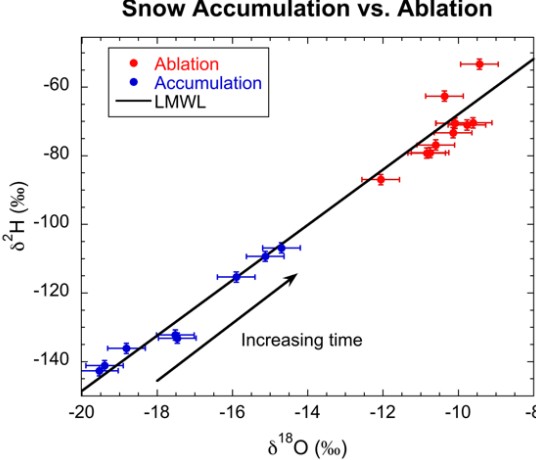

**Figure 4: $\delta^2H$ and $\delta^{18}O$ measurements from the snow accumulation (blue filled circles) and snow ablation (red filled circles) period**
**as well as the local meteoric water line (LMWL) represented by the solid black line.**

The enrichment of $^{18}O$ and $^2H$ during the snow accumulation period can also not be attributed to sporadic rain events. Rain
that falls during the accumulation period has a lighter isotopic signature compared to snow since the formation of snow from
air moisture leads to a higher enrichment of heavy isotopes compared to the formation of rain (Clark and Fritz, 1997). Hence,





the enrichment of $^{18}$O and $^{2}$H in the snow during the accumulation period can only be related to the refreeze of meltwater
and/or the moisture exchange with the underlying Earth surface or the atmosphere since these processes enrich the snow in
$^{18}$O and $^{2}$H isotopes along the LMWL (Beria et al., 2018; Steen-Larsen et al., 2014). Compared to the snow accumulation
period, more enriched $\delta^{18}$O and $\delta^{2}$H average values (-10.34‰; -72.1‰) were measured during the ablation period (May to
October) (Fig. 2), which is in agreement with previous studies (Dietermann and Weiler, 2013; Lee et al., 2010; Zhou et al.,
2008). Similar to the accumulation period, no significant deviation from the LMWL was observed during the snow ablation
period, revealing that sublimation processes were not the predominant isotope fractionation process (Fig. 4). The more enriched
$\delta^{18}$O and $\delta^{2}$H snow values in the ablation compared to the ablation period can be likely explained by the contribution of rain,
which has a heavier isotopic signature compared to the snow during the ablation period. Besides, similar to the accumulation
period, the refreezing of meltwater and the exchange with the Earth surface and atmosphere could also contribute to the
enrichment of heavy oxygen ($^{18}$O) and hydrogen ($^{2}$H) isotopes during the snow ablation period.

288        For determining the $\delta^{18}$O and $\delta^{2}$H values of the glacial ice end-member both the solid and the melting ice was sampled.

Similar to the snow samples, no significant aberration of the $\delta^{18}$O and $\delta^{2}$H values from the LMWL was detected for both the
solid and melting ice (Fig. 5). This indicates that also for the glacial ice, sublimation processes played a minor role and that
the $\delta^{18}$O and $\delta^{2}$H glacial ice signatures were primarily controlled by melting/refreezing processes and the contribution of rain
water and moisture. Compared to the solid ice ($\delta^{18}$O = -14.12‰ and $\delta^{2}$H = -101.7‰), the melted ice showed slightly more
enriched $\delta^{18}$O and $\delta^{2}$H values showing a shift of $\Delta\delta^{18}$O = 1.02‰ and $\Delta\delta^{2}$H = 7.2‰, respectively in average (Fig. 5).
Additionally, the average $\delta^{18}$O and $\delta^{2}$H values (-13.42‰; -96.9‰) of the ice (solid and melting) were higher compared to the
snow in the accumulation period and more depleted compared to the snow in the ablation period and the rain samples (Fig. 2).
The somewhat intermediate glacial $\delta^{18}$O and $\delta^{2}$H values compared to the snow in the ablation and accumulation period is
plausible since the glacial ice is formed from snow that originates from both the ablation and accumulation period (Beria et
al., 2018).

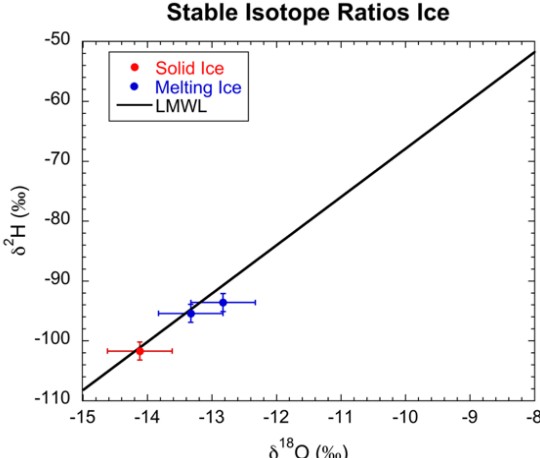

**Figure 5: $\delta^{2}$H and $\delta^{18}$O measurements from the melting (blue filled circles) and solid ice (red filled circles) as well as the local meteoric water line (LMWL) represented by the solid black line.**

### 3.2 Qualitative discharge separation based on temporal stream analysis in the three catchment areas

To evaluate the contribution of the different end-members (snow, ice and rain) to the stream discharge in the three catchments,
highly temporally resolved hydro-chemical analysis were conducted in the effluents of the three catchments between June/July
2019 and March 2020 in the Giglibach and Steinwasser catchment and between August 2019 and March 2020 in the





Wendenwasser catchment. The hydro-chemical measurements included discharge volumes, electrical conductivity and stable
water isotope ratios ($\delta^{18}$O and $\delta^2$H) as well as precipitation (Fig. 6).

**Figure 6: Temporal evolution of the precipitation in the study area (A) as well as discharge (B), electrical conductivity E.C. (C), stable oxygen (D), and stable hydrogen isotope ratios (E) measurements in the effluents of the Giglibach (blue circles), Steinwasser (orange) and Wendenwasser (grey circle) catchments between June 2019 and March 2020. The temporal precipitation evolution (A) was acquired from the precipitation measuring station in Gadmen operated by the Office of Waste and Water (AWA) of the Canton of Bern, Switzerland.**



The stream discharges in the three investigated catchments were highest in the Steinwasser followed by the
Wendenwasser and Giglibach catchments (Fig. 6B) correlating to the different sizes of the catchments (Fig. 1). The stream
discharges in the Steinwasser and Wendenwasser catchments showed large temporal variations and can be divided into a high
(June – August 2019), intermediate (September – October 2019) and low discharge time period (November – March 2020)
(Fig. 6B). During the high discharge period, the Steinwasser and Wendenwasser stream discharges ranged between 3 and 10
$m^3/s$ and between 1 and 3 $m^3/s$, respectively, with a few peak discharges of up to 12 $m^3/s$ and 9 $m^3/s$, respectively during heavy
precipitation events (Figs. 6A and B). The intermediate discharge period was dominated by short discharges peaks in both the
Stein- and Wendenwasser catchment (up to 9 $m^3/s$), which were also related to precipitation events (Fig. 6A) followed by
baseflow recessions to discharges of around 0.80 $m^3/s$ (Figs. 6A and B).  At the beginning of the high discharge phase (June –
mid-July 2019), only discharge data for the Steinwasser catchment is available. During this time, the stream discharge in the
Steinwasser catchment is likely dominated by the snow melt, since the snow cover is rapidly decreasing during this time period
(Fig. 3), which is consistent with previous observations and simulations (Hydro-CH2018). In the second half of the high (mid-
July – August 2019) and in the intermediate discharge phase (September – October 2019) discharge data for both the Stein-
and Wendenwasser catchment is available. During this time period the snow cover has disappeared (Fig. 3) and the glacial
meltwater becomes most probably the main contributor to the stream discharges in the Stein- and Wendenwasser catchment.
The low $\delta^{18}O$ and $\delta^2H$ values (~ -12‰; -85‰) in the stream discharges of the Stein- and Wendenwasser catchments further
support the significant snow and glacial melt water stream discharge contribution between mid-July and October 2019. The
$\delta^{18}O$ and $\delta^2H$ values are close to the snow and ice end-member values and higher $\delta^{18}O$ and $\delta^2H$ values closer to the rain end-
member (~ -11‰; -77‰) are only observed during heavy precipitation events (Figs. 6A, D and E). The significant contribution
of snow and glacial meltwater to the stream discharges is further reinforced by the low electrical conductivity (E.C.) in the
Steinwasser catchment discharge (~ 30 μs/cm) between June and August 2019 (Fig. 6C) since snow and glacial meltwater
usually show a lower E.C. compared to the rain water contribution via surface run-off (Krainer and Mostler, 2002; Zuecco et
al., 2019).
In the low discharge period in the Stein- and Wendenwasser catchment, the discharge was less high compared to the
high and intermediate discharge phase (Fig. 6B). However, the discharge measurements were associated with high
uncertainties due to the partial freeze of the measuring stations and the missing calibration measurements. In contrast to the
high and intermediate discharge phase, the Stein- and Wendenwasser catchments stream discharges during the wintry baseflow
period (November 2019 - March 2020) were likely controlled by groundwater inflow into the streams representing a mixture
between the rain, ice and snow end members. Hence, during this time period it is challenging to identify the relative
contribution of the different end-members (ice, rain, snow) to the total catchment discharges also because no isotope data of
the rain end-member is available for this time period.
As opposed to the Stein- and Wendenwasser catchment, a lower temporal discharge variation was observed in the
Giglibach catchment showing an average discharge of 0.8 $m^3/s$ between July and October 2019 and even lower discharges
between November 2019 and March 2020 during the wintry baseflow. Similar to the Stein- and Wendenwasser catchment
effluents, the determination of the wintry baseflow in the Giglibach catchment was associated with uncertainties due to the
partial freeze of the measuring station and the missing calibration measurements. The overall lower temporal discharge
variation in the Giglibach catchment can be explained by the lower average altitude and hence, by the lower contribution of
snow melt water during summer, causing strong seasonal variations of the discharge as observed in the Stein- and
Wendenwasser catchment. Nevertheless, also in the Giglibach catchment discharge, depleted $\delta^{18}O$ and $\delta^2H$ values close to the
isotopic signature of snow and ice were measured between July and October 2019. This indicates that also in the Giglibach
catchment the snow and glacial meltwater significantly contributes to the discharge between July and October 2019.





**3.3 Quantitative discharge separation based on stable isotope ratio in the catchment's effluents**

The quantitative discharge separation in the three catchments was conducted based on the stable water isotopes measurements and focused on the glacial meltwater contribution to the catchments' effluents between August and September 2019. This time period is of special interest since the glacial meltwater contribution to the stream discharges is a) likely highest throughout the year due to the combination of high temperatures and the absence of snowmelt and b) subject to disappearance in the future due to climate change-induced deglaciation. Therefore, the quantification of the glacial meltwater contribution in August and September is crucial to predict discharges of mountainous streams in future in the course of climate change. The absence of snowmelt in September and August also provides the advantage that only two end-members (rain and ice) need to be taken into account for the quantification of the glacial melt water contribution using stable water isotopes, which facilitates the data interpretation. The glacial meltwater contribution between August and September 2019 was quantified using the temporal stable isotope ratio measurements ($\delta^{18}O$ and $\delta^2H$) in the catchment effluents (Fig. 6D and E) and the determined isotopic signature of the end-members (Figs. 2 and 5) using equation 2. For the rain end-member, the temporal variation of the $\delta^{18}O$ and $\delta^2H$ values in each of the catchment outlets was taken into account, while the altitude effect was not considered as no overall altitude correction factor could be established (Fig. 2). For the $\delta^{18}O$ and $\delta^2H$ signature of the melting ice end-member, constant $\delta^{18}O$ and $\delta^2H$ values representing the average of the two taken samples ($\delta^{18}O$ = -13.08‰ and $\delta^2H$ = -94.5‰; Fig. 5) were used for the discharge separation calculations as it can be expected that the isotopic signature of the melting ice changes minimally between in August and September (Beria et al., 2018). The uncertainty of determined glacial melt water contribution was evaluated based on the uncertainty of the stable isotope measurement in the stream discharges and the contributing end-members combined with the Gaussian error propagation law, which was applied to equation 2. This resulted in an uncertainty of the determined glacial melt water contribution of ±2%.

The quantitative evaluation of the stream discharge for the three catchments in August and September 2019 revealed a high glacial melt contribution of up to 95±2% (Fig. 7) and is likely restricted to these two months. The relative discharge

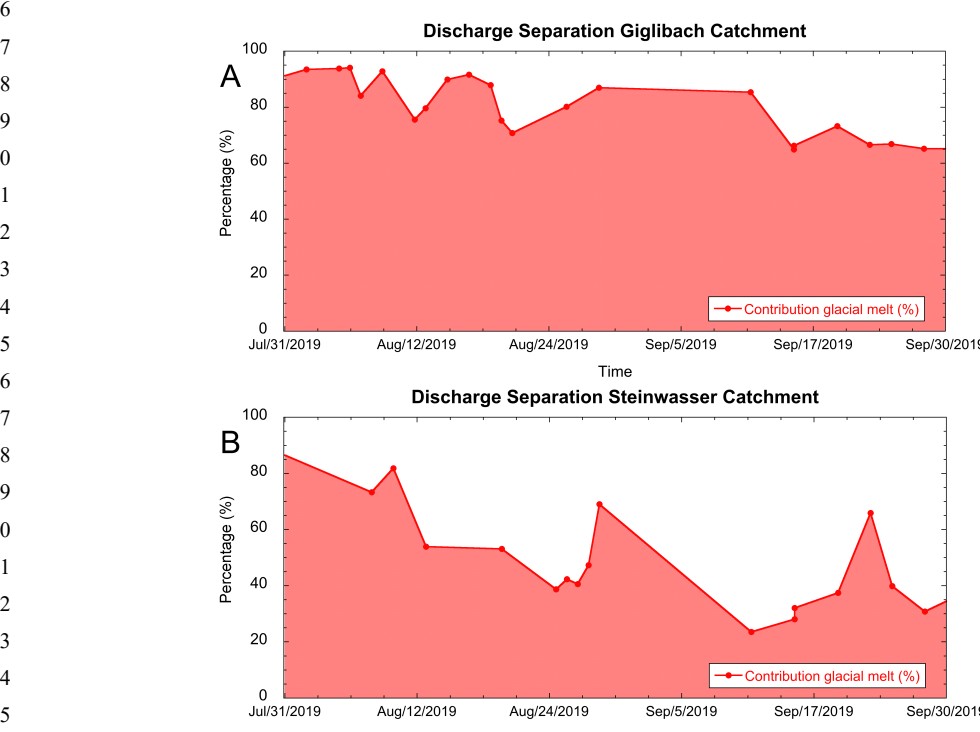
















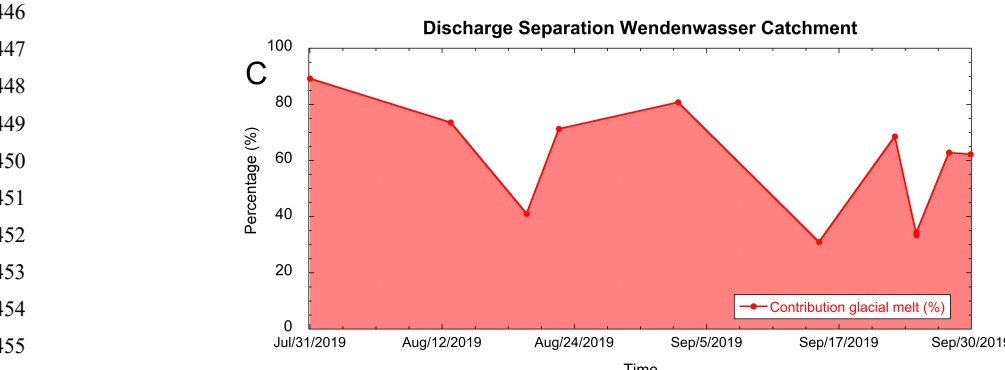

**Figure 7: Quantification of glacial melt water contribution the three catchment discharges in August and September 2019 using**
**stable isotope ratios measurements ($\delta^{18}$O and $\delta^{2}$H) in the catchment effluents and the end-members using equation 2. The uncertainty**
**of the determined glacial meltwater contribution corresponds to ±2%.**

contribution of the glacial melt water was slightly higher in the Giglibach catchment compared to the Stein- and Wendenwasser
catchment (Fig. 7). This can be explained by the lower altitude of the glaciated area and the related higher temperatures leading
to a higher melting rate of the glaciated areas of the Giglibach catchment compared to the Steinwasser and Wendenwasser
catchment. The relative glacial meltwater contribution to the stream discharges also varied over time in August and September
2019. The highest glacial melt contribution to the stream discharges was observed at the beginning of August, reaching values
between 95±2 and 80±2% in the Steinwasser and Giglibach catchment, respectively, and above 80% in the Wendenwasser
catchment stream discharge (Fig. 7). In late August and early September, the relative glacial melt water discharge contribution
to the stream discharges was slightly lower compared to early August, but still above 70% in the Giglibach and Wendenwasser
catchment and above 40% in the Steinwasser catchment. Towards the end of September, the relative glacial meltwater
contribution to the catchments' discharges further decreased but not below 50% in the Giglibach catchment and not below
20% and 30% in the Steinwasser and Wendenwasser catchment, respectively (Fig. 7). The high relative glacial melt water
contribution to the stream discharges in three catchments in August and September shows that the stream discharges will likely
decrease in the future in August and September, due to the climate change-induced deglaciation.
The determined relative glacial meltwater water contribution to the stream discharges (Fig. 7) can be further used to
estimate the minimum annual glacial meltwater discharge volume (mAGMD) for the three catchments given the assumption
that the glacial meltwater mostly contributes to the stream discharges in August and September. These calculations can be
made by multiplying the relative glacial meltwater water contributions (Fig. 7) by the measured total discharges (Fig. 6B) and
by integrating them over time between August and September. This resulted in mAGMDs of 3.5 Mio m$^3$ for the Giglibach,
17.9 Mio m$^3$ for the Steinwasser, and 9.6 Mio m$^3$ for the Wendenwasser catchment cumulating in a total annual mAGMD of
31.0 Mio m$^3$ for all three catchments. By including the uncertainty of the relative meltwater contribution calculations and the
total discharge measurements, the uncertainty associated with the determination of the mAGMDs is 10%. When plotting these
mAGMDs versus the glaciated areas in the three catchments as well as the zero-point representing a non-glaciated catchment
with zero mAGMD, a power-law relation between the mAGMD and the glaciated area can be observed (Fig. 8). This relation
shows that the mAGMD is proportionally higher for smaller compared to larger glaciated areas. This is plausible since smaller
glaciated areas are exposed to a larger extent to warm air compared to larger glaciated area leading to proportionally higher
mAGMDs. Furthermore, the smaller glaciers are usually located at a lower altitude compared to large glaciers, where
temperature are higher also contributing to the proportionally higher mAGMDs for smaller compared to larger glaciated areas.
The detected relationship (Fig. 8) between the mAGMD and the catchment's glaciated areas provides the advantage that it can
used for other mountainous catchments at similar altitudes to estimate the glacial meltwater discharge volume based on the





catchment's glaciated area only. This facilitates the estimation of the annual glacial melt water discharge volumes since the
glaciated area of a catchment is easier to determine than conducting temporal stable water isotope measurements in streams
and in the contributing end-members.

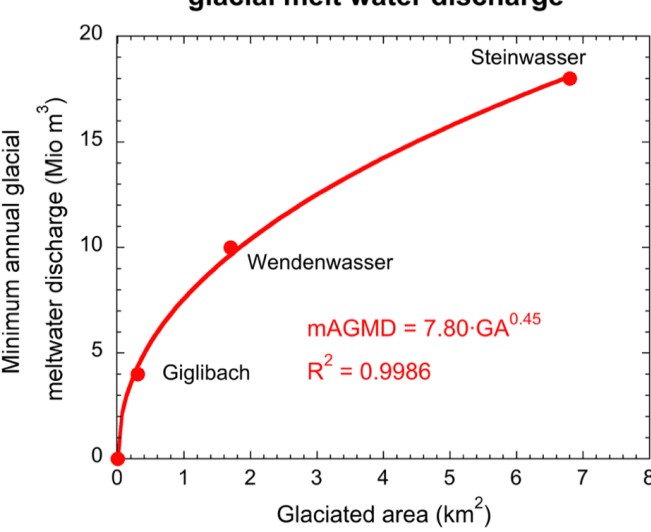

**Figure 8: Power-law relation between the glaciated area (GA) and the minimum annual glacial meltwater discharge volumes**
**(mAGMD) in Mio m³ in the three in catchments (Giglibach, Wendenwasser, Steinwasser).**
The determined mAGMD can also be used to quantify the contribution of the glacial melt water discharge to the total annual
mountainous stream discharge in three catchments. The total annual mountainous stream discharges for the three catchments
can be estimated based on discharge simulations by the Federal Office for the Environment of Switzerland (Pfaundler and
Schönenberger, 2013). These simulations revealed annual discharge volumes of 62.4 Mio m³ for the Steinwasser, 31.2 Mio m³
for the Wendenwasser, and of 14.2 Mio m³ for the Giglibach catchment, respectively, resulting in a total annual discharge
volume for the three catchments of 107.8 Mio m³. These simulated annual discharge volumes were highly consistent with
internal annual stream discharge volume measurements in the three catchments in 2019 by the Kraftwerke Oberhasli AG
(personal communication) reinforcing the robustness and the representativity of these annual stream discharge volumes for the
year 2019. The relation of the total annual discharge volumes to the mAGMD for the three catchments results in annual glacial
melt water discharge contributions of 24.5%±5% for the Giglibach, 28.7%±5% for the Steinwasser, and 30.7%±5% for the
Wendenwasser catchment and of 28.7%±5% for all three catchments together. These relatively high annual glacial meltwater
discharge contribution in the three catchments reinforces the hypothesis that discharge regimes in mountainous catchments
will change in the future when these glacial meltwater contributions will cease caused by climate change.
**4. Conclusions**
The stream discharge separation in three partially glaciated alpine catchments based on stable water isotope measurements
revealed a high contribution of glacial meltwater of up to 95±2% in August and September, corresponding to a glacial
meltwater contribution to the total annual discharges of 28.7±5%. It is expected that these high glacial meltwater contributions
to mountainous stream discharges will decrease not only in our study area but also in other Alpine regions during the next
decades due to global warming-induced deglaciation. Moreover, the peak discharges in the mountainous streams will likely





occur earlier in the year (May/June) compared to today (June/July) due to the earlier occurrence of the snowmelt caused by
global warming. However, predictive discharge simulations for Alpine catchments suggest that the annual mountainous stream
discharge volumes will not significantly decrease despite of the ceasing glacial meltwater contributions as they will be
compensated by higher discharge volumes in winter and spring (Hydro-CH2018). Nevertheless, the changing flow regimes in
mountains streams caused by climate change will pose a challenge for hydropower energy production in Alpine regions. Hence,
the operation of hydropower energy production using artificially dammed lakes in alpine regions needs to adapt to these
changing flow regimes in mountainous streams caused by climate change. This is major importance for achieving a carbon
neutral energy production, as hydropower energy is the most important renewable energy resource and it is crucial that
hydropower energy is exploitable to the same extent in the future despite global warming to further reduce greenhouse gas
emissions.
Overall, this study demonstrates a successful monitoring strategy for three partially glaciated mountainous catchments
for quantifying the glacial meltwater contribution to stream discharges based on stable water isotope measurements. In
particular, the study showed that for a successful quantification of the glacial meltwater contribution based on stable water
isotopes a high temporal resolution of the end-members and catchment discharges is necessary, especially of snow and rain as
they vary strongly over time. Moreover, our results showed that a quantification of the glacial meltwater contribution is only
possible when snow meltwater is absent as the isotopic signature of snow and rain overlap. However, this is no major drawback
since the glacial meltwater contribution is only significant when no snowmelt is occurring. Additionally, the annual glacial
meltwater discharge volumes in three catchments showed an excellent power-law correlation with the catchment's glaciated
area. This correlation allows the estimation of the annual glacial meltwater discharge volume in other mountainous catchments
based on the glaciated area only. This is an advantage as the glaciated area is easier to determine than stable water isotope
measurements in mountainous streams and in the contributing end-members. Taken as whole, an implementation of the
developed sampling strategy in this study to other mountainous catchments will provide an improved validation of existing
mountainous catchment modelling studies for the quantification of the glacial meltwater contribution to streams in
mountainous regions.

**5. Data availability**

The raw the data for this study can be accessed in the Zenodo data repository through: https://doi.org/10.5281/zenodo.5571465

**6. Author contribution**

Philipp Wanner: Conceptualization, methodology, analytical laboratory work, writing – original draft preparation. Noemi Buri:
Field work, writing – review & editing. Kevin Wyss: Field work, writing – review & editing. Andreas Zischg:
Conceptualization, methodology, writing – review & editing. Rolf Weingartner: Methodology, writing – review & editing. Jan
Baumgartner: Methodology, writing – review & editing. Benjamin Berger: Methodology, field work, writing – review &
editing. Christoph Wanner: Conceptualization, methodology, writing – review & editing.

**7. Competing interest**

The authors declare that they have no conflict of interest

**8. Acknowledgements**

The authors acknowledge the great technical support from the Kraftwerke Oberhasli AG (KWO) during the sampling campaign
in three partially glaciated Alpine catchments.



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
