# Peer review of "Quantifying the glacial meltwater contribution to streams in mountainous regions using highly resolved stable water isotope measurements"

_Hydrology and Earth System Sciences, 2021_

## Referee Comment (RC1)

Review of hess-2021-512, by Philipp Wanner, Noemi Buri, Kevin Wyss, Andreas Zischg, Rolf Weingartner, Jan Baumgartner, Benjamin Berger, and Christoph Wanner, entitled: Quantifying the glacial meltwater contribution to streams in mountainous regions using highly resolved stable water isotope measurements

**By Bettina Schaefli**

This paper proposes to quantify the contribution of ice melt to total streamflow in three highly glaciated catchments in the central Swiss Alps with the help of astable isotopes of water. The aim is to come up with results that are more reliable than previous modelling-based results and with recommendations for future sampling campaigns.

I cannot recommend the paper for publication because some fundamental hydrological process knowledge is ignored. The obtained results are not plausible (glacier melt contribution of between 80% and 95% to total streamflow during August in catchments with only between 6 and 28% glacier cover). One key result is summarized in Figure 8, which shows glacier melt in Mio m$^3$ against glacier area. For the smallest glacier investigated, this result indicates meltwater production of $4*10^6$ m$^3$ on an area of 0.3 km$^2$, which corresponds to a melt water production of $4*10^6$ m$^3$/$0.3*10^6$ m$^2$ = 13.3 m of melt water production over the glacier area. For the largest glacier, it is $18*10^6$ m$^3$/$6.8*10^6$ m$^2$ = 2.7 m of meltwater production. The first value is impossible, the last values is in the order of observed summer mass balances in Switzerland in 2019 (see Figure 1).

[Figure]

[Figure]

*Figure 1: Observed summer mass balances in Switzerland from (GLAMOS, 2020), all values since 2011 (left) and for summer 2019 (right)*

The reasons for the erroneous estimates are certainly related to the wrong assumption that streamflow during summer is only composed of glacier melt and of rainfall. In reality, an important part of streamflow is groundwater (baseflow) released by the hillslopes; the isotopic values of groundwater are strongly influenced by snow melt and thus close to the values of glacier melt (see below). Accordingly, the separation into glacier melt and not-glacier melt is impossible with the help of isotopes alone. EC values could help separating ground water from non-groundwater input but this would require values for groundwater and values for ice melt at the glacier snout (which was already in contact with the ground).

**Detailed comments**

The analysis of the contribution of ice to streamflow is based on a total of 2 ice melt samples taken each from a different glacier, both located in only one of the three catchments, i.e. there are no ice samples in two of the catchments. One catchment has no snow samples, all snow samples have (according to the sampling location figure) been taken at low elevations, there is only a total of 19 snow samples (the paper does not contain a clear overview of dates and elevations when and where the snow samples were taken).

Ice melt can have considerable variability (Figure 2) and be overlapping with the values of snowmelt and of the snowpack (Figure 3). Since groundwater is strongly influenced by snowmelt, it most likely has isotopic ratios that are also rather low.

[Figure]

*Figure 2: delta-Deuterium values in ice melt samples from the Otemma glacier (Müller et al., 2021), see also the display material here: https://presentations.copernicus.org/EGU21/EGU21-7182_presentation.pdf*

[Figure]

*Figure 3: delta-Deuterium values in ice melt and snow melt samples from the Otemma glacier (Müller et al., 2021)*

**Additional specific comments**

- Introduction: There is no reference to mixing analysis in the introduction despite of the huge body of hydrologic literature in this field. There is very little reference to isotopes studies in Alpine areas (e.g. Penna et al., 2014)
- Line 252 following: The text mentions the enrichment in heavy isotopes in the snowpack over the accumulation season and attributes it to melt/refreeze cycles and moisture exchange with the ground. This explanation is a priori not plausible for enrichment during the accumulation phase

at elevations around 2000 masl (exact sampling elevations unknown) where ground is often frozen in winter and melt only occasion. However, the sampled period might well correspond to an exceptionally warm winter. This should be specified. We would need actual temperature recordings to shed light on this.

- Line 283: mistake, "The more enriched d18O and d2H snow values in the *ablation* compared to the *ablation* period"
- Line 288: I would not interpret a single solid ice sample with respect to two ice melt samples.
- Line 363 following: would be more interesting to compare the streamflow in terms of specific discharge (normalized to catchment area), in mm/d, (and thus remove the log-scale in the figure)
- Line 381: "The significant contribution of snow and glacial meltwater to the stream discharges is further reinforced by the low electrical conductivity (E.C.) in the Steinwasser catchment discharge (~ 30 µs/cm) between June and August 2019 (Fig. 6C)": you omit that the two others seem to have values of around 100. Do you have any groundwater / spring sample to judge how high this is?
- Line 389 following: you make the point that during winter low flow, which is dominated by groundwater, the separation of streamflow components (rain, snow, ice) is difficult. This applies also during the rest of the year
- Line 405 following: do you have evidence of the absence of snow in August and September? Perhaps at least the largest glacier has still a firn / permanent snow area? Even for the other two glaciers, snow might persists in August and might come back in late September? Complete absence might hold maximum for a week or two. Snow might even persist in August in shady areas outside the glaciers?
- Line 415: you could test the sensitivity of the results to a lapse rate in precipitation, since you have such an effect for part of the year as far as I understood?
- Line 420: "it can be expected that the isotopic signature of the melting ice changes minimally between in August and September (Beria et al., 2018)." Different locations on the glacier might show different values for melt; but the actual problem is that the hillslopes provide high baseflow, which has isotopic values of groundwater, which in turn has the values of snow;
- Line 424: your main result with very high glacier melt shares for all three catchments is not in-line with your EC measurements?
- Figure 8: fitting a power-law to three points is clearly over-fitting
-

**References**

GLAMOS: Swiss Glacier Mass Balance, release 2020, Glacier Monitoring Switzerland, 10.18750/massbalance.2020.r2020., 2020.

Müller, T., Schaefli, B., and Lane, S. N.: Assessing the effect of the geomorphological complexity of glacier forefields on the multi-temporal water dynamics will provide better future models, EGU General Assembly 2021, EGU21-7182, 10.5194/egusphere-egu21-7182, 2021.

Penna, D., Engel, M., Mao, L., Dell'Agnese, A., Bertoldi, G., and Comiti, F.: Tracer-based analysis of spatial and temporal variations of water sources in a glacierized catchment, Hydrol. Earth Syst. Sci., 18, 5271-5288, 10.5194/hess-18-5271-2014, 2014.

---

## Author Comment (AC1)

**Author responses to comments Reviewer #1 (Review of hess-2021-512)**

This paper proposes to quantify the contribution of ice melt to total streamflow in three highly glaciated catchments in the central Swiss Alps with the help of stable isotopes of water. The aim is to come up with results that are more reliable than previous modelling-based results and with recommendations for future sampling campaigns.

1. I cannot recommend the paper for publication because some fundamental hydrological process knowledge is ignored. The obtained results are not plausible (glacier melt contribution of between 80% and 95% to total streamflow during August in catchments with only between 6 and 28% glacier cover). One key result is summarized in Figure 8, which shows glacier melt in Mio m3 against glacier area. For the smallest glacier investigated, this result indicates meltwater production of $4*10^6$ m3 on an area of 0.3 km2, which corresponds to a melt water production of $4*10^6$ m3/$0.3*10^6$ m2 = 13.3 m of melt water production over the glacier area. For the largest glacier, it is $18*10^6$ m3/$6.8*10^6$ m2 = 2.7 m of meltwater production. The first value is impossible, the last value is in the order of observed summer mass balances in Switzerland in 2019 (see Figure 1 one in the attached complete review).

We thank the reviewer for this highly important estimation and we acknowledge that we should have done this plausibility calculation by ourselves. However, in our opinion, these estimations also illustrate that our quantification of the glacial meltwater contribution worked for the catchment with the highest degree of glaciation (Steinwasser). Moreover, the high variability of the electrical conductivity (EC) data in the Steinwasser catchment compared to the other two catchments (Fig. 6 of the original manuscript) demonstrates that the relative groundwater contribution to the mountainous streams the is much lower compared to the other two catchments. Consequently, the glacial meltwater production estimated by Reviewer #1 for the Steinwasser catchments yielded reasonable results because the groundwater contribution is low compared to the other catchments. This sets an important limitation for using stable water isotope data to quantify glacial meltwater contributions to mountainous streams such that the stable isotope method works if the groundwater contribution is low. We plan to highlight this in the revised version of the manuscript if we are allowed to revise the manuscript.

2. The reasons for the erroneous estimates are certainly related to the wrong assumption that streamflow during summer is only composed of glacier melt and of rainfall. In reality, an important part of streamflow is groundwater (baseflow) released by the hillslopes; the isotopic values of groundwater are strongly influenced by snow melt and thus close to the values of glacier melt (see below). Accordingly, the separation into glacier melt and not-glacier melt is impossible with the help of isotopes alone. EC values could help separating ground water from non-groundwater input but this would require values for groundwater and values for ice melt at the glacier snout (which was already in contact with the ground).

As discussed in the general response to the reviewer's comments, we agree that our partially erroneous estimates of the glacial meltwater contribution to mountainous streams are related to the negligence of the groundwater as an interim storage for all end-members (snowmelt, glacial melt, rainwater). Also, we agree that EC values are crucial for identifying a significant groundwater contribution to the streamwater samples. For instance, the high EC variation observed for the Steinwasser catchment (Fig. 6) is likely inherited from a lower groundwater contribution compared to the Wendenwasser and Gigli catchments showing

much lower EC variations typical of groundwater-dominated streams. However, groundwater is not an independent end-member because it consists of a mixture of the three endmembers (snowmelt, glacial melt, rainwater). Thus, in our opinion, quantifying the groundwater component does not significantly help to get better estimates for the contribution of the rainwater, snowmelt, and glacial melt contribution in a specific streamwater sample. This is also because the EC value of the groundwater component is mainly controlled by the overall degree of mineral dissolution reactions occurring in the subsurface and thus, the subsurface residence time of the groundwater body, and not by the contribution of the three end-members in a specific groundwater sample.

3. The analysis of the contribution of ice to streamflow is based on a total of 2 ice melt samples taken each from a different glacier, both located in only one of the three catchments, i.e. there are no ice samples in two of the catchments. One catchment has no snow samples, all snow samples have (according to the sampling location figure) been taken at low elevations, there is only a total of 19 snow samples (the paper does not contain a clear overview of dates and elevations when and where the snow samples were taken). Ice melt can have considerable variability (Figure 2) and be overlapping with the values of snowmelt and of the snowpack (Figure 3). Since groundwater is strongly influenced by snowmelt, it most likely has isotopic ratios that are also rather low.

In our opinion, the low number of glacial meltwater samples does not majorly affect the main conclusion of the paper since it has been previously shown that the isotopic variability of glacial melt water is low compared to snow and rain (Müller et al., 2021; Schmieder et al., 2018; Zuecco et al., 2019). In addition, the data that reviewer 1 shows in figure 2 displays a $\delta^2$H variation of 6‰ (-106 - -112‰) for glacial meltwater. A similar $\delta^2$H variation of was observed in our samples (-93.6 - -95.3‰). Moreover, the variation in $\delta^2$H in figure 2 shown by the reviewer corresponds to a variation of around 0.75‰ in $\delta^{18}$O being also in the range of our samples (-12.83‰ vs. -13.33‰). Hence, we think that we captured the isotopic variability of the glacial meltwater with our samples and that they are representative for the glacial meltwater in the mountainous streams.

Regarding the 23 snow samples, it is correct that they originate from only one catchment. However, given that the outlets of the three catchments are located within a distance of only about 2 km (Fig. 1 of the original manuscript), we do not think that this is an issue. The sampling dates and exact coordinates are all provided in the data repository (see line 557 of the original manuscript). However, we agree that we should provide at least the sampling altitudes in a table of the main manuscript. This will demonstrate that the snow samples were collected at different altitudes ranging from 1541 to 2169 masl. We would also like to emphasize that we have collected the snow samples during a period of 13 months (February 2019-March 2020), which included the winter months where the catchments were only accessible by helicopters. We are confident that the resulting snow stable isotope data presented in Fig. 4 represent a unique dataset for obtaining new insights into the processes causing the temporal variation of the stable isotope signature of alpine snow packs. For the revised manuscript, we plan to provide a more in-depth interpretation and discussion of this highly interesting and unique dataset if we are allowed to revise the manuscript.

We also agree that groundwater is influenced by snowmelt but we are not convinced that snowmelt is the only source. Instead, groundwater likely represents a mixture between snowmelt, rainwater and glacial melt. Owing to the elevated residence time we expect that

groundwater $\delta^2$H and $\delta^{18}$O values reflect an average signature defined by the mean annual contribution of snowmelt, glacial melt and rainfall as the three main water sources.

4. Figure 2: delta-Deuterium values in ice melt samples from the Otemma glacier (Müller et al., 2021), see also the display material here: https://presentations.copernicus.org/EGU21/EGU21- 7182_presentation.pdf

Unfortunately, the indicated link does not work so we could not find the indicated material.

5. Introduction: There is no reference to mixing analysis in the introduction despite of the huge body of hydrologic literature in this field. There is very little reference to isotopes studies in Alpine areas (e.g. Penna et al., 2014)

We acknowledge that we have to extend the introduction regarding this topic. This will be done in the revised version of the manuscript if we are allowed to revise the manuscript.

6. The text mentions the enrichment in heavy isotopes in the snowpack over the accumulation season and attributes it to melt/refreeze cycles and moisture exchange with the ground. This explanation is a priori not plausible for enrichment during the accumulation phase at elevations around 2000 masl (exact sampling elevations unknown) where ground is often frozen in winter and melt only occasion. However, the sampled period might well correspond to an exceptionally warm winter. This should be specified. We would need actual temperature recordings to shed light on this.

We agree that moisture exchange with the subsurface is not possible and that moisture exchange with the atmosphere is more likely to have caused the observed stable isotope shift. In the revised manuscript, we will provide an extended interpretation and discussion of the data shown in Fig. 6 if we are allowed to revise the manuscript.

The exact sampling locations are provided in the data repository and we will list the sampling altitudes in the main part of the revised version of the manuscript if we are allowed to revise the manuscript.

7. Line 283: mistake, "The more enriched d18O and d2H snow values in the ablation compared to the ablation period".

Correct, the second time "ablation" should be replaced by "accumulation". We will correct this issue in the revised manuscript if we are allowed to revise the manuscript.

8. Line 288: I would not interpret a single solid ice sample with respect to two ice melt samples.

Well, the samples were taken at the same time and we think it is an interesting observation because it demonstrates that during ice melt, stable isotope ratios can change. We agree, that the interpretation is challenging though.

9. Line 363 following: would be more interesting to compare the streamflow in terms of specific discharge (normalized to catchment area), in mm/d, (and thus remove the log-scale in the figure)

We agree with the reviewer and we will apply the suggested change when preparing the revised version of the manuscript if we are allowed to revise the manuscript.

10. Line 381: "The significant contribution of snow and glacial meltwater to the stream discharges is further reinforced by the low electrical conductivity (E.C.) in the Steinwasser

catchment discharge (~ 30 µs/cm) between June and August 2019 (Fig. 6C)": you omit that the two others seem to have values of around 100. Do you have any groundwater / spring sample to judge how high this is?

In fact, we did collect spring samples (i.e. groundwater) close to the outlet of the Wenden- and Steinwasser catchments. For the Wendenwasser spring, we measured 148 and 149 µS/cm on August 23 and September 16, 2019, respectively. For the Steinwasser spring, we measured 63 µS/cm on October 3 2019. These measurements are fully consistent with the statement the reviewer referred to (line 381) and we plan to add them to the manuscript and to provide a corresponding discussion when preparing the revised version of the manuscript if we are allowed to revise the manuscript.

11. Line 389 following: you make the point that during winter low flow, which is dominated by groundwater, the separation of streamflow components (rain, snow, ice) is difficult. This applies also during the rest of the year

We agree with the reviewer. As described in the general response to the reviewer's comments, we plan to change slightly the scope of the manuscript to focus more on the opportunities, challenges, and limitations of using stable water isotopes for the quantification of glacial meltwater contributions to mountainous streams.

12. Line 405 following: do you have evidence of the absence of snow in August and September? Perhaps at least the largest glacier has still a firn / permanent snow area? Even for the other two glaciers, snow might persists in August and might come back in late September? Complete absence might hold maximum for a week or two. Snow might even persist in August in shady areas outside the glaciers?

The reviewer is right, we cannot completely exclude the presence of firn and the absence of patchy snow in shady areas. However, the observation that our quantification approach results in reasonable glacial meltwater contributions to the stream in the Steinwasser catchment, characterized by a low groundwater contribution (see general response to the reviewer's comments above), suggest that the error introduced by this simplification is rather small.

13. Line 415: you could test the sensitivity of the results to a lapse rate in precipitation, since you have such an effect for part of the year as far as I understood?

The reviewer is right, this could and will be tested when preparing the revised manuscript if we are allowed to revise the manuscript.

14. Line 420: "it can be expected that the isotopic signature of the melting ice changes minimally between in August and September (Beria et al., 2018)." Different locations on the glacier might show different values for melt; but the actual problem is that the hillslopes provide high baseflow, which has isotopic values of groundwater, which in turn has the values of snow;

We agree, as described in the general response to the reviewer's comments, the quantification of the glacial meltwater contribution to mountainous streams is highly challenging if a strong groundwater contribution occurs. However, we would like to emphasize that the stable isotope values of groundwater samples depend on the contribution of snowmelt, glacial melt and rainfall in the groundwater. Owing to the elevated residence time we expect that groundwater $\delta^2$H and $\delta^{18}$O values reflect an average signature

defined by the mean annual contribution of snowmelt, glacial melt and rainfall as the three main water sources.

15. Line 424: your main result with very high glacier melt shares for all three catchments is not in-line with your EC measurements?

We agree with the reviewer. As mentioned in the general response to the reviewer's comments, we acknowledge that only the glacial meltwater contribution to the streams for the Steinwasser catchment is plausible.

16. Figure 8: fitting a power-law to three points is clearly over-fitting?

As described in in the general response to the reviewer's comments, we plan to slightly shift the scope of the manuscript and we will no longer provide fully quantitative estimates of glacial meltwater contribution for the Giglibach and the Wendenwasser catchment. Therefore, Figure 8 will be removed when preparing the revised version of the manuscript if we are allowed to revise the manuscript.

**References**

Müller T., Schaefli B. and Lane S. N. (2021) Assessing the effect of the geomorphological complexity of glacier forefields on the multi‑temporal water dynamics will provide better future models. *EGU General Assembly 2021, EGU21‑7182, 10.5194/egusphere‑egu21‑7182, 2021*.

Schmieder J., Garvelmann J., Marke T. and Strasser U. (2018) Spatio-temporal tracer variability in the glacier melt end-member — How does it affect hydrograph separation results? *Hydrological Processes* **32**, 1828-1843.

Schmieder J., Hanzer F., Marke T., Garvelmann J., Warscher M., Kunstmann H. and Strasser U. (2016) The importance of snowmelt spatiotemporal variability for isotope-based hydrograph separation in a high-elevation catchment. *Hydrol. Earth Syst. Sci.* **20**, 5015-5033.

Zuecco G., Carturan L., De Blasi F., Seppi R., Zanoner T., Penna D., Borga M., Carton A. and Dalla Fontana G. (2019) Understanding hydrological processes in glacierized catchments: Evidence and implications of highly variable isotopic and electrical conductivity data. *Hydrological Processes* **33**, 816-832.

---

## Author Comment (AC2)

**Author responses to Reviewer #2 ('Comment on hess-2021-512)**

1. The authors of this manuscript analyzed the temporal variability in the isotopic composition of rain water and snow samples, and quantified the contribution of glacial melt water to stream runoff, by means of stable water isotopes, in three study catchments in the Swiss Alps.

The topic of this manuscript is potentially interesting for the readers of Hydrology and Earth System Sciences. In general, I think that more studies investigating the contribution of snowmelt and glacier melt to stream runoff in high elevation catchments are needed to improve our understanding of hydrological processes in such complex areas. Overall, the paper is well structured and well written, but I have several (major) concerns about the methodological approach.

Firstly, the authors have not considered the contribution of groundwater to runoff both in the accumulation and the ablation period. Groundwater is expected to be the dominant end-member during the accumulation period, but a large contribution of groundwater to runoff may be possible from the glacier-free areas of the catchments during the ablation period.

Secondly, more details are needed in the section 2.4 about the hydrograph separation. The authors should explain the choice of the end members, provide the assumptions at the base of the hydrograph separation technique (please see Klaus and McDonnell, 2013), and describe how uncertainty was estimated (it is mentioned only at lines 420-423).

We thank Reviewer #2 for his/her important comments. As mentioned in the general response to the reviewer's comments, we acknowledge that our approach of neglecting groundwater as significant interim storage for glacial melt, rainwater, and snowmelt was somewhat simplified and that our dataset does not allow making strong quantitative statements regarding the glacial meltwater contribution to mountainous streams for all hydrological set-ups. Therefore, we plan to slightly shift the scope of the manuscript and we intend to focus more on the opportunities, challenges, and limitations of using stable water isotopes to quantify the contribution of glacial meltwater to mountainous streams. This will also include a more detailed uncertainty analysis of the hydrograph separation in the monitored stream discharges and the corresponding isotope signatures if we are allowed to revise the manuscript.

2. Thirdly, the authors should consider more and discuss the temporal and spatial variability in the isotopic composition of the end members. Previous studies conducted in Alpine catchments (e.g., Schmieder et al., 2016; Schmieder et al., 2018; Zuecco et al., 2019) have already shown that a high spatial and temporal variability in the tracer composition of the end members can greatly affect the results of the hydrograph separation and/or hamper its application. In this study, the authors used only three samples of glacier ice (and from only one of the glaciers) to characterize the glacier-melt end member. This sample size is too small for making any consideration on hydrograph separation.

In our opinion the strength of our manuscript is the isotopic analysis of the snow samples collected during a consecutive period of 13 months (February 2019 -March 2020), whereby during the winter months the samples could only be collected via the usage of helicopters. Moreover, a high number samples was also collected in the streams at the outlet of the three catchments over a 10 months period. We think that the collected snow and stream samples in our study provide additional information to previous studies (e.g. Müller et al., 2021;

Schmieder et al., 2016; Zuecco et al., 2019) such that they provide new insights into the processes causing the temporal variation of the stable isotope signature of alpine snow packs (Fig. 4). For this reason, we plan to provide an extended discussion regarding this important topic in the revised version of the manuscript if we are allowed to revise the manuscript.

Regarding the glacier-melt endmember, the reviewer is right that the number of samples is low. Nevertheless, since it has been previously shown that the isotopic variability of glacial melt is rather low, at least when compared to that of snow (Müller et al., 2021; Schmieder et al., 2018; Zuecco et al., 2019), Hence, we think that we captured the isotopic variability of the glacial meltwater with our samples and that they are representative for the glacial meltwater in the mountainous streams.

3. Finally, the authors have not described which approach was used to assess the end of the snowmelt period in the three catchments (using snow cover data collected at only one station at 2063 m a.s.l. is not sufficient).

We agree with the reviewer that it is challenging to define the end of the snowmelt period. We chose the end of July because this is roughly one month after the snow cover disappeared at the snow cover measurement station (Fig. 3). Thus, for catchments with subsurface residence times of less than a month, most of the snowmelt that was stored in the subsurface has left the catchment. The observation that the quantified meltwater generation for August to September yields reasonable values for the Steinwasser catchment (see the general response to the reviewer's comments) suggests that the chosen approach including the choice of the end of the snowmelt period is reasonable. Our quantification approach provides reasonable estimates either because the groundwater contribution during this period was low or because the corresponding subsurface residence time is short (in the order of less than month). This discussion will be added to the revised version of the manuscript if we are allowed to revise the manuscript.

4. The introduction is mainly focused on the role of hydropower in Alpine catchments, whereas there is too little attention towards the application of tracers in high-elevation catchments to quantify the contribution of glacier-melt water to stream runoff."

We agree that the introduction was biased towards the role of hydropower. Since we plan to slightly change the scope of the manuscript, the introduction will be adapted according to the revised scope of the manuscript if we are allowed to revise the manuscript.

5. Lines 47-48: This concept repeats the text at lines 32-35

Thanks for indicating that. We plan to update the introduction such that it will be in line with the anticipated shifted scope of the manuscript if we are allowed to revise the manuscript.

6. Line 54: I would not describe the tracer-based methods as low cost compared to other methods, such as hydrological modeling.

We agree that tracer-based methods are labor-intensive, particularly because of the remote and hardly accessible terrain. Thus, we will remove the statement that the tracer-based methods are low-cost methods in the revised manuscript if we are allowed to revise the manuscript.

7. In the legend of Figure 1, I suggest indicating the glacierized area.

In Figure 1, the glaciated areas are shown in lighter colors than the rest of the three catchments. This will be described in the updated figure caption in the revised manuscript if we are allowed to revise the manuscript.

8. Line 123: 19 snow samples is not a high sample size.

We see the point of the reviewer and we will remove this characterization of the snow sample number. Instead, we will emphasize that the snow samples were collected during more than an entire year (February 2019 – March 2020), which is one of the main novelty of our sampling campaign.

9. Line 131: I suggest indicating the number of ice samples that were collected.

We agree with this suggestion and we will indicate the number of the collected ice samples in the revised manuscript if we are allowed to revise the manuscript.

10. Lines 132-133: Three samples collected at the glacier fronts cannot be representative of the whole ablation zone. Additional samples are needed to support the main findings of this manuscript.

See the response to comment 2.

11. Lines 274-276: These two sentences are not supported by rain samples collected during the accumulation period.

In Figure 2, we did not report any rainwater isotopic data for the accumulation period. Therefore, we do not fully understand what the reviewer means here.

12. Lines 363-371 and Figure 6: I suggest comparing discharge values after normalization by catchment areas.

We agree with the reviewer and we will apply the suggested change when preparing the revised version of the manuscript if we are allowed to revise the manuscript.

13. Lines 410-411: The author should provide evidence about the presence/absence of snowmelt in all three catchments during the ablation period.

In our monitoring period, we recorded two minor snowfall events in early September and mid October 2019 (<10 cm). The fresh snow has quickly molten on the next day. At the end of the ablation period on November 10, 2019 (Fig. 3), no snow was present in the three catchments based on webcam monitoring of the area at the Susten Pass at an altitude of 2224 masl. We will add this information to the revised manuscript if we are allowed to revise the manuscript.

14. Lines 420-423: These sentences belong to section 2.4.

We agree that these sentences should be part of the method section. We plan to move them to the method section when preparing the revised manuscript if we are allowed to revise the manuscript.

15. Figure 8: This figure could be interesting if more catchments were considered; is it possible to gather data from other Alpine catchments? If not, I suggest deleting the figure

As described in in the general response to the reviewer's comments, we plan to slightly change the scope of the manuscript and we will no longer provide fully quantitative estimates of glacial meltwater contribution for the Giglibach and the Wendenwasser catchment discharge. Therefore, Figure 8 will be removed when preparing the revised version of the

manuscript if we are allowed to revise the manuscript.

16. Line 172: It is unclear what the authors mean with "binary mixing approach". I suggest using another term, such as "two-component hydrograph separation"

We used a binary mixing model with two end-members and this is the reason why the term "binary mixing approach" was used. However, we have no issues with changing the term as suggested. This will be done when preparing the revised manuscript if we are allowed to revise the manuscript.

17. Line 223: Please indicate the water source for "heavy isotopes".

The source relates to rainwater, which condensates and precipitates at lower altitude. This information will be added in brackets when preparing the revised manuscript if we are allowed to revise the manuscript.

18. Lines 225-226: Please mention the water source considered in the sentence

The water source refers to rainwater and this information will be added to the sentence when preparing the revised manuscript if we are allowed to revise the manuscript.

19. Figure 4: Please indicate in the caption what the error bars represent.

The error bars represent the analytical uncertainty of $\delta^2H$ and $\delta^{18}O$ as described in the method section. For $\delta^{18}O$ it is 0.10 ‰, for $\delta^2H$ it is 1.5‰. This information will be added when preparing the revised manuscript if we are allowed to revise the manuscript.

20. Figure 5: Please indicate in the caption what the error bars represent.

The error bars represent the analytical uncertainty of $\delta^2H$ and $\delta^{18}O$ as described in the method section. For $\delta^{18}O$ it is 0.10 ‰, for $\delta^2H$ it is 1.5‰. This information will be added when preparing the revised manuscript if we are allowed to revise the manuscript.

**References**

Müller T., Schaefli B. and Lane S. N. (2021) Assessing the effect of the geomorphological complexity of glacier forefields on the multi‑temporal water dynamics will provide better future models. *EGU General Assembly 2021, EGU21‑7182, 10.5194/egusphere‑egu21‑7182, 2021.*

Schmieder J., Garvelmann J., Marke T. and Strasser U. (2018) Spatio-temporal tracer variability in the glacier melt end-member — How does it affect hydrograph separation results? *Hydrological Processes* **32**, 1828-1843.

Schmieder J., Hanzer F., Marke T., Garvelmann J., Warscher M., Kunstmann H. and Strasser U. (2016) The importance of snowmelt spatiotemporal variability for isotope-based hydrograph separation in a high-elevation catchment. *Hydrol. Earth Syst. Sci.* **20**, 5015-5033.

Zuecco G., Carturan L., De Blasi F., Seppi R., Zanoner T., Penna D., Borga M., Carton A. and Dalla Fontana G. (2019) Understanding hydrological processes in glacierized catchments: Evidence and implications of highly variable isotopic and electrical conductivity data. *Hydrological Processes* **33**, 816-832.

---

## Author Comment (AC3)

**Author responses to Reviewer #3 (Comment on hess-2021-512)**

1. This article estimates the role of glacial meltwater in generating stream discharge in three Alpine catchments located in the Central Swiss Alps. Stable water isotopes (2H, 18O) are used to quantify the proportion of streamflow generated from ice melt vs rainfall while electrical conductivity measurements are qualitatively used to understand the dominant hydrologic processes. The article concludes that ice melt is the dominant driver of streamflow generation in August and September and propose that due to climate change, glacial coverage will reduce which might adversely impact streamflow generation during this period of the year. The article then estimates annual glacial melt discharge in these three catchments and propose a power law relationship between minimum annual glacial meltwater discharge and the glaciated area, which can potentially be extrapolated to catchments with known glaciated areas.

The paper is well written but lacks significantly in terms of robustness of the methods used and the inferences made thereafter. The key problem that I see is one missing end-member which is "groundwater" that has not been considered in this article. In Alpine environments, groundwater has a significant role is sustaining streamflow during low flow periods in August-October period. In this particular case study, I think groundwater is significantly contributing to streamwater generation during August-September period as can be inferred from the high EC values during that part of the year (Figure 6C). If this period was completely dominated by ice melt originating from glaciers, EC values would be much lower and similar to that observed in the June-July period in Steinwasser catchment when snowmelt was dominating streamwater recharge (Figure 6C). As Steinwasser is the only catchment which has a longer timeseries of EC values, we can see that snowmelt was probably dominating stream recharge in June, July (low EC values) and then groundwater kicked-in in late August which is why EC values increased significantly. As the article has only relied on stable isotope measurements, this distinction is missing. I want to see if the results would be similar if the end member mixing exercise was undertaken with EC values and not stable water isotopic ratios. This also makes sense because electrical conductivity is largely a conservative tracer.

We thank reviewer#3 for these thorough comments. As mentioned in the general response to the reviewer's comments, we agree that our approach of neglecting groundwater as significant interim storage for glacial melt, rainwater, and snowmelt was somewhat simplified and that our dataset does not allow making strong quantitative statements regarding the glacial meltwater contribution to mountainous streams. Hence, we plan to slightly change the focus of the manuscript and we plan to address more the opportunities, challenges, and limitations of using stable water isotopes to quantify the contribution of glacial meltwater to mountainous streams

In addition, we agree that EC values are crucial to identify an important groundwater contribution to streamwater samples such as suggested by Reviewer #3. However, we somewhat disagree that the EC is a conservative tracer. This is because EC values are proportional to the sum of solutes dissolved in the groundwater. Solute concentrations in turn are mainly controlled by the overall degree of mineral dissolution reactions occurring in the subsurface, which for granitic systems is directly proportional to the subsurface residence time of the groundwater. Thus, because of this reactive behavior, groundwater EC values cannot be used to determine the contribution of rainwater, snowmelt and glacial melt in specific groundwater samples. Instead, EC values of streamwater samples can be used to quantify the corresponding groundwater component. However, this does not necessarily help in getting

better estimates for the contributions of the rainwater, snowmelt and glacial melt endmembers in a specific streamwater sample because groundwater consists of the same three end-members.

2. In terms of mechanism, I think there might be significant subsurface storage that is getting recharged by snowmelt and ice melt (hence very depleted) and this storage is then recharging the stream during August September period. If this mechanism is indeed true, then the underlying hypothesis that rapidly retreating glaciers will lead to very low flows in August September period will not be true as groundwater can be recharged via rainfall, snowmelt and ice melt. I would like to hear the authors' perspective on this and if this was considered as a possible hypothesis.

Well, if the glacial meltwater contribution decreases then the discharge is much more sensitive to meteorological variations because they control the amount of snow and rainfall that is recharging and is stored in the groundwater systems. For instance, for dry years, the discharge will be much lower after the disappearance of the glaciers, whereas for wet years the lacking glacial meltwater contribution might be compensate by the contribution of stored groundwater originating from rain and snow melt water. This discussion will be added to the revised manuscript if we are allowed to revise the manuscript.

3. Variability in the isotopic ratio of ice melt (originating from the glacier) is very low and might not be very realistic. This is probably due to very limited ice sampling (only sampled two times in August and September, L418). Hence, the distinction in isotopic ratio of ice melt and snowmelt might be more of a function of sampling bias rather than any underlying hydrologic process.

We agree with the reviewer that the number of glacial melt samples is low. Nevertheless, since it has been previously shown that the isotopic variability of glacial melt is rather low, at least when compared to that of snow (Müller et al., 2021; Schmieder et al., 2018; Zuecco et al., 2019), Hence, we think that we captured the isotopic variability of the glacial meltwater with our samples and that they are representative for the glacial meltwater in the mountainous streams.

4. L521-523: I find it very surprising that the ice melt contributes to ~25% of total discharge in Giglibach when the extent of glacial coverage is only 8%. On the other hand, the extent of glacial coverage is as high as 28% in Steinwasser but the contribution of glacial melt to total discharge is only slightly higher at ~29%. Are these estimates reasonable or to put it differently, have these kinds of number been reported at any other place where despite very high glacial coverage (>3x for Steinwasser compared to Giglibach), contribution to annual stream discharge only increases slightly.

As nicely pointed out by Reviewer #1, our estimations for the Giglibach catchment cannot be true. The erroneous quantification results from neglecting the groundwater contribution, which is much higher in the Giglibach catchment compared to the other catchments (see also general response to the reviewer's comments).

5. L377: Groundwater might also be a significant contributor to stream recharge. I propose the authors to explore this hypothesis

We agree with the reviewer. An extended discussion regarding groundwater will be added when preparing the revisions if we are allowed to revise the manuscript (see also general response to the reviewer's comments).

6. L381-385: If snow and glacial meltwater show lower EC compared, then August and September discharge cannot be explained by glacial meltwater as EC values are high across catchments

Theoretically, the August and September discharge could be related to glacial meltwater that has interacted with the minerals in the subsurface and thereby increased its solute concentrations and hence, EC values. In any case, groundwater flow is significant (but variable in the different catchments) and we fully acknowledge that we have to provide a corresponding discussion in the revised manuscript if we are allowed to revise the manuscript (see also general response to the reviewer's comments).

7. L418: Two samples is very few to make any meaningful statistical judgement

See the response to comment 3.

8. L420-423: Details about Gaussian error propagation has not been explained anywhere in the article. Additionally, ±2% uncertainty bound seems to be very small. This might be due to small sample size.

We agree that an uncertainty of ±2% is too low. This is mainly because we have neglected groundwater as a significant interim storage for rainwater, snowmelt and glacial melt. An extended discussion regarding the corresponding uncertainty for quantifying the glacial meltwater contribution will be added when preparing the revised manuscript.

9. L483-486: Has this been reported for the first time? I am not familiar with this literature, are there other studies which have reported similar results? In that case, it might be good to include relevant references.

As described in in the general response to the reviewer's comments, we plan to slightly change the scope of the manuscript and we will no longer provide fully quantitative estimates of glacial meltwater contribution for the Giglibach and the Wendenwasser catchment discharge. Therefore, Figure 8 will be removed when preparing the revised version of the manuscript if we are allowed to revise the manuscript.

10. L544-545: Using temporally high-resolution isotope measurements leading to superior quantification of glacial meltwater hasn't been shown in this article

Temporally high resolution refers to snow and streamwater samples as pointed out in earlier replies. This will be clarified in the revised manuscript if we are allowed to revise the manuscript. In addition, we will delete the somewhat misleading statement that our data has led to an improved quantification of glacial meltwater contributions. Instead, we will emphasize that our dataset provides new insights into the opportunities, challenges and limitations of using stable water isotopes for such quantification.

11. L284: It should read as "… in the ablation compared to the accumulation period …"

Thank you for spotting this error. We will correct it accordingly in the revised manuscript if we are allowed to revise the manuscript.

12. L25: It might be clearer if its written as "… which has a heavier isotopic signature compared to the snow that fell during the accumulation period…"

Thank you for this nice suggestion. We will correct it accordingly in the revised manuscript if we are allowed to revise the manuscript.

13. L538: Should be ". This is of major importance .."

Thank you for spotting this error. We will correct it accordingly in the revised manuscript if we are allowed to revise the manuscript.

14. Figure 1: Incorrect figure caption, Wendenwasser is shown in grey and not pink.

Thank you for spotting this error. We will correct it accordingly in the revised manuscript if we are allowed to revise the manuscript.

15. Figure 5: Should also include snowmelt isotopic ratios here to make the comparison between snowmelt and ice melt easier. Is this any reason to believe that both will have different isotopic signature?

We agree that it makes sense to merge Figure 4 and 5. This will make it clearer that the isotopic snow signature is strongly variable and hence, overlapping with the signature of ice. The variation is of the snow signature is likely caused by the moisture exchange with the atmosphere after snowfall. An extended discussion regarding this process will be added to the revised manuscript if we are allowed to revise the manuscript.

16. Figure 6: In subplots B, C and D there is a lot of whitespace due to very large y-axis bounds. For e.g. there are no discharge measurements below 0.1 m3/s, so showing y-axis values up to 0.01 m3/s is not necessary. Similar is the case for EC values < 10. I will suggest the authors to consider using tighter y-axis bounds so that the underlying data variability is more clearly visible.

Thank for this suggestion, we will certainly consider it when preparing a revised version of the manuscript if we are allowed to revise the manuscript.

17. Figure 6A: Is the unit mm or mm/hr?

Fig. 6A shows daily amounts of precipitation. The unit should thus be mm/d. This will be clarified in the revised manuscript if we are allowed to revise the manuscript.

18. Figure 7: I will suggest adding uncertainty bounds in this figure. Also, is 90%+ glacial melt contribution (Figure 7A) a plausible estimate at the end of July in a catchment which is only 6% glaciated?

As pointed out earlier, the estimates shown in Fig. 7a cannot be entirely correct because the Giglibach catchment is likely strongly affected by groundwater flow, which we have neglected as interim water storage. Adding uncertainty bounds is a good suggestion and we will consider this when revising the manuscript if we are allowed to revise the manuscript.

19. Figure 7 caption: Should be ".. glacial melt water contribution to the three .."

Thank you for spotting this error. We will correct it accordingly in the revised manuscript if we are allowed to revise the manuscript.

**References**

Müller T., Schaefli B. and Lane S. N. (2021) Assessing the effect of the geomorphological complexity of glacier forefields on the multi‑temporal water dynamics will provide better future models. *EGU General Assembly 2021, EGU21‑7182, 10.5194/egusphere‑egu21‑7182, 2021*.

Schmieder J., Garvelmann J., Marke T. and Strasser U. (2018) Spatio-temporal tracer variability in the glacier melt end-member — How does it affect hydrograph separation results? *Hydrological Processes* **32**, 1828-1843.

Zuecco G., Carturan L., De Blasi F., Seppi R., Zanoner T., Penna D., Borga M., Carton A. and Dalla Fontana G. (2019) Understanding hydrological processes in glacierized catchments: Evidence and implications of highly variable isotopic and electrical conductivity data. *Hydrological Processes* **33**, 816-832.